# Beyond the Barrier: Targeted Radionuclide Therapy in Brain Tumors and Metastases

**DOI:** 10.3390/pharmaceutics11080376

**Published:** 2019-08-01

**Authors:** Janik Puttemans, Tony Lahoutte, Matthias D’Huyvetter, Nick Devoogdt

**Affiliations:** 1In Vivo Cellular and Molecular Imaging Lab, Vrije Universiteit Brussel, Laarbeeklaan 103, 1090 Brussels, Belgium; 2Nuclear Medicine Department, UZ Brussel, Laarbeeklaan 101, 1090 Brussels, Belgium

**Keywords:** brain cancer, brain metastasis, radionuclide therapy

## Abstract

Brain tumors are notoriously difficult to treat. The blood-brain barrier provides a sanctuary site where residual and metastatic cancer cells can evade most therapeutic modalities. The delicate nature of the brain further complicates the decision of eliminating as much tumorous tissue as possible while protecting healthy tissue. Despite recent advances in immunotherapy, radiotherapy and systemic treatments, prognosis of newly diagnosed patients remains dismal, and recurrence is still a universal problem. Several strategies are now under preclinical and clinical investigation to optimize delivery and maximize the cytotoxic potential of pharmaceuticals with regards to brain tumors. This review provides an overview of targeted radionuclide therapy approaches for the treatment of primary brain tumors and brain metastases, with an emphasis on biological targeting moieties that specifically target key biomarkers involved in cancer development.

## 1. Introduction

Malignant brain tumors remain to this day among the most lethal of all cancer types and is the 7th most common cancer in the United States (US). With no cure available for grade III-IV gliomas and almost universal recurrence, diagnosis is still a death sentence, with a 5-year survival rate ranging between 5% to 35% [1]. Primary brain tumors progress from cells with a cerebral origin, whereas secondary brain tumors have spread and seeded into the brain from the primary tumor site elsewhere in the body. With an incidence of 22 per 100.000, primary brain tumors account for about 2% of all cancers, of which gliomas—cancer cells that originate from glial precursors—represent 75% of all malignant primary brain tumors in adults. Recurrent high-grade gliomas are associated with a median overall survival (OS) of less than one year, and all treatments are palliative and associated with severe side effects [1,2]. 

Systemic tumors often have the ability to invade into nearby tissues, progress into regional lymph nodes, or metastasize to other organs. Metastatic or secondary brain tumors arise as a consequence of circulating cancer cells nesting within the brain. Metastasis to the central nervous system is an indicator of poor prognosis and is nearly always fatal. The global incidence of brain metastases in patients with systemic cancer is probably around 10 times higher than that of primary brain tumors, and as many as 10%–30% of patients with metastatic cancer will develop brain metastases [3,4,5]. The most common primary tumors that metastasize to the brain are lung cancer (19.9%), melanoma (6.9%), renal cancer (6.5%), breast cancer (5.1%) and colorectal cancer (1.8%) [4]. 

## 2. The Problem of the Brain

Since the advent of the modern struggle against cancer, surgery, chemotherapy, and radiotherapy remain the cornerstone for disease management. While overall survival has been improved significantly for many cancer types, a major hurdle remains management of tumors localized within the brain. The occurrence of brain metastasis is even expected to increase, due to better management of extracranial disease and longer survival of patients. 

Surgical debulking is recommended for large, or strategically located tumors, but is often hindered by the invasive nature of glial tumors into the surrounding tissue. Removal of difficult-to-reach lesions can also endanger vital cerebral structures. These are so-called inoperable tumors and require either radiation therapy or systemically administered pharmaceutics. 

External beam radiotherapy is the most commonly applied treatment option and can be provided under the form of radiosurgery for single or limited number of tumor sites, or whole-brain radiotherapy (WBRT) for patients with multiple lesions where focal treatments are ineffective. Despite the efficacy of WBRT, median survival prolongation is modest at best and often accompanied by transient worsening of neurological symptoms, radiation-induced necrosis, and dementia [6]. 

Brachytherapy is a method of irradiating tumors by placing an encapsulated radioactive source inside the body of the patient next to or within the tumor tissue. This allows the use of higher total doses of radiation to treat a smaller area. As a result, this greatly reduces the risk of exposing healthy tissues to cytotoxic radiation. Brachytherapy is also limited to treating single localized, easily accessible lesions and is therefore most often applied for low-grade gliomas, anaplastic astrocytomas, and meningiomas [7]. 

The value of chemotherapy is still under some debate. The therapeutic efficacy of cytotoxic drugs is greatly dependent on chemosensitivity of the cancer type, the vascularization of the tumor, and the degree of permeability of the blood-brain-barrier (BBB). The BBB comprises tight junction and adherence proteins in between endothelial cells, transmembrane transporters, basal lamina, and extracellular matrix. Not only do tight junction and adherence proteins inhibit paracellular diffusion of hydrophilic molecules from the blood to the CNS; several efflux transporters in the BBB also export potentially harmful endogenous and exogenous substances from the CNS back to the blood circulation [8]. The natural impermeability of the BBB and the strictly regulated homeostasis in the brain have shown to be major limitations for chemotherapeutics to reach cancerous lesions. Since the introduction of the alkylating agent temozolomide (TMZ) in 1997 [9], little advancement has been made in the field of CNS-appropriate chemotherapeutics, meaning patients with unmanageable intracranial metastases often have to resort to the same cytotoxic chemotherapy that is utilized for the treatment of extracranial disease.

Taking into account the collateral damage near the tumor margin inflicted during surgical resection to ensure maximal debulking, the exposure of healthy brain tissue during WBRT, and the lack of specificity of chemotherapeutics, there has been a developmental shift towards more tumor-specific strategies that limit off-target toxicity. Improved knowledge of cancer biology has led to the discovery of certain molecular drivers of cancer development, progression and metastasis. The (over)expression of these tumor markers has opened the field to developing targeted therapies to specifically trigger cancer cell death or arrest cell growth. Novel strategies include immunotherapy, targeted therapy, and gene therapy. Immunotherapy has made great strides for controlling systemic lesions, but lags when it comes to intracranial disease, mostly due to the immune-privileged nature of the CNS, caused by cytotoxic T-cell exhaustion, limited lymphocytic infiltration, recruitment of protumorigenic tumor-associated macrophages, downregulation of cancer cells’ MHC I complexes, and an abundance of immune-inhibitory molecules such as IL-10 and transforming growth factor β. [10,11,12,13]. This impasse has led to a growth in the development of tyrosine kinase inhibitors and tumor-targeting moieties such as monoclonal antibodies and antibody fragments, which are nanoparticles that do not necessarily depend on an activated immune system for eradication.

Unfortunately, the activation of alternate molecular-genetic pathways during treatment renders the modus operandi of several targeting agents ineffective, leading to pathway inhibition therapy resistance [14,15,16]. A possible way to circumvent this acquired resistance is to not only rely on the inherent anti-tumoral effect of the targeting vehicle, but also to equip it with a cytotoxic payload, as has been the case for antibody-drug conjugates (ADC), e.g., Trastuzumab emtansine (Kadcyla^TM^). This can either be a biologically active cytotoxic drug, or a cytotoxic radionuclide. A potential downside of using cytotoxic drugs is that they need to be internalized by the cancer cell and intracellularly released before they can perform their biological function on quickly dividing cells only [17], whereas utilizing targeted radionuclide therapy (TRNT) allows this radiopharmaceutical to exert its anti-tumor effect when bound to the cell surface by inducing DNA damage through radioactive decay. Depending on the travel distance of the emitted cytotoxic particle, more than one cancer cell can be targeted per decay event. This is called the crossfire effect [18]. This can aid in tumor killing, but might also lead to off-target toxicity when healthy tissues are irradiated [19]. Coupling a potent radionuclide to a selective targeting moiety in order to kill receptor-expressing cancer cells could result in a highly effective therapeutic approach that combines the non-invasive nature, ease-of-administration, and systemic distribution of chemotherapeutics with the efficacy of locally applied external beam radiation without the off-target toxicity. As an added advantage of TRNT, radiopharmaceuticals can be utilized as single-photon emission computerized tomography (SPECT)- or positron emission tomography (PET)-radiotracers by exchanging the cytotoxic radionuclide with a gamma- or positron-emitting radionuclide resp., or by using therapeutic radionuclides that also emit γ-radiation. This concept of using the same targeting moiety for diagnostic and therapeutic use, is generally referred to as ‘theranostics’. 

## 3. Matching the Vehicle to the Emitter

The biological impact of the radiopharmaceutical is determined by the type of ionizing radiation of the implemented radionuclide. Whereas gamma (γ)- and positron (β^+^)-emitting radionuclides are ideal for molecular nuclear imaging, due to the strong penetration of gamma rays through matter and relatively low ionization potential, they are not fit for radionuclide therapy. One class of radionuclides are those that emit Auger electrons. Electrons that are emitted via the Auger effect are large in numbers, yet low in kinetic energy, which results in a short travel path of only a few nanometers—less than the size of a single cell. For this reason, Auger-based radiopharmaceuticals must be internalized by the cancer cell in order to exert their anti-tumoral effect. 

More appropriate therapeutic radionuclides emit either beta- (β^−^)- or alpha (α)-particles. Until recently, TRNT has been applied mainly using β^−^-particle-emitting radionuclides. β^−^-particles are electrons emitted from an unstable atom nucleus due to an excess of neutrons. These electrons can travel up to a few millimeters through tissue but possess a relatively low linear energy transfer (LET) (~0.2 keV/μm). Due to low LET, the induced damage can consist of either irreparable double- or mostly repairable single-DNA strand breaks, which could result in sublethal damage. On the other hand, in an oxygen-rich environment, β^−^-radiation can cause indirect cell-killing via the formation of reactive oxygen species (ROS), adding to its cytotoxic effect. The formation of ROS by β^−^-particles combined with their long travel path, makes them a preferred choice for bulky, heterogeneous tumors. 

For small tumors, micrometastatic lesions, or residual disease after surgery, α-particles—which are in fact emitted helium-4 nuclei—are a potentially better alternative, owing to their short travel distance in tissue (only a few cell diameters) and high LET (50–230 keV/μm). These characteristics make them potent ionization agents with greater biological effectiveness than either conventional external beam radiation or β^−^-emitters. Theoretically, a cancer cell can be killed by a few α-particle hits, and its efficacy is independent of cell cycle phase or the formation of free oxygen radicals, meaning that even slowly dividing and hypoxic tumors resp. can be eliminated with targeted alpha therapy (TAT) [20]. An overview of the most commonly used therapeutic radionuclides is given in Table 1.

The efficacy of TRNT is based on specific accumulation of the radiolabeled tracer in cancerous tissue, while being rapidly cleared from non-target organs. For the highest possible tumor uptake, it is important for the targeting vehicle to have a very high affinity for its target, combined with deep tissue penetration. In order to ensure fast clearance from healthy tissue, a small vector is preferable. The problem that arises is that large molecules—such as monoclonal antibodies—provide the highest tumor accumulation, whereas smaller molecules—such as peptides—provide the most favorable tumor-to-normal organ dose ratios [21].

To benefit from the biomarker-driven approach to deliver ionizing radiation to cancer cells without potential off-target toxicities, it is important to match the biological half-life of the targeting vehicle with the radioactive half-life of the radionuclide. Longer circulating vectors require the use of long-lived isotopes, which also implies longer exposure of healthy tissues. However, short circulating vectors can carry a radioactive payload with a longer half-life, as it is the goal for the accumulated fraction in the cancer lesions to remain for multiple days or even weeks. Ultimately, it is the delivered radiation dose over time inside the cancer lesions that will determine the success of the treatment. Despite the availability of different molecular vectors, most targeted radionuclide therapies still utilize monoclonal antibodies to deliver the radionuclide to the tumor site. In order to bypass the strictly regulated passage through the BBB of these large, hydrophilic molecules and increase tumor uptake, several strategies are implemented. The most commonly applied strategy is to administer the radiopharmaceutical as a bolus directly into the resection cavity after tumor debulking [22,23,24]. This serves the purpose of eliminating residual cells that are responsible for eventual recurrence, usually within 2 cm of the tumor margins, as well as reducing possible systemic toxicity caused by intravenous, intra-arterial or intrathecal administration.

Instead of relying on passive diffusion of the targeting moiety through the dense brain tissue, convection-enhanced delivery (CED) can be used to increase local delivery of therapeutics to the tumor site. CED is a drug delivery method that uses a motor-driven pumping device connected to a catheter tip that is stereotactically placed at the target site within the brain. The hydraulic pressure in CED allows for a homogeneous distribution of large molecules by displacing interstitial fluid with the drug-containing infusate [25,26,27]. 

When local application of CED or surgical resection is impossible due to critical lesion location, systemic delivery of radiopharmaceuticals can still be improved by transiently opening the tight junctions enclosing the BBB. Different approaches exist to induce transient BBB disruption. Intracarotid or intravenous infusion of hypertonic solutions such as mannitol causes shrinkage of endothelial cells, resulting in a physical passage in between these cells [28,29]. Interestingly, despite the severe risks of hypertonic infusions, such as electrolyte abnormalities, hypotension, and renal and cardiac dysfunction, mannitol is routinely administered to patients undergoing debulking brain tumor surgery in order to reduce intracranial pressure. However, there is no standardized dosage of mannitol for brain relaxation, which impedes interpretation of new clinical studies when compared to historical controls [30].

A more specific BBB disruptive agent is RMP-7 (Cereport^®^). RMP-7 is a synthetic bradykinin analogue that induces selective permeability of the BBB through transient relaxation of tight junctions, and thus permeating cerebrovasculature. RMP-7 has mainly been applied to increase delivery of carboplatin in glial tumors, however with little efficiency and phase III clinical trials with RMP-7 were discontinued [31,32,33]. The efficacy of RMP-7 in combination with TRNT or in context of brain metastases has not been clinically investigated. 

Alternatively, focused ultrasound-mediated BBB disruption—where intravenously administered microbubbles oscillate due to high-energy ultrasound waves, causing tight junctions in the BBB to open—is a promising technique but requires prior knowledge of all metastatic sites [34].

Considering the large size of monoclonal antibodies (~150 kDa), there has been a developmental shift towards the use of smaller antibody fragments, such as minibodies (80 kDa), diabodies (55 kDa), single-chain variable fragments (scFv) (25 kDa), and heavy-chain-only antibody fragments (VHH) (15 kDa) [35,36,37]. These engineered fragments potentially offer better BBB passage, tissue penetration, and their shorter blood circulation half-life implies the use of shorter-lived radionuclides. Their fast clearance from the blood and healthy tissue also means a high tumor-to-background ratio soon after administration, making them more suitable as theranostic tracers. However, despite their implementation as diagnostic tracers, none of these constructs has made it to clinical testing with regards to therapeutic use in the context of brain tumors. 

## 4. Clinical Applications

Chapter 4 provides an overview of past and current TRNT approaches that focus on brain-related cancers. A condensed overview of all clinical advancements is summarized in Table 2. 

### 4.1. Peptide Receptor Radionuclide Therapy

#### 4.1.1. Somatostatin Receptors

Peptide receptor radionuclide therapy (PRRT) allows systemic treatment of cancer types that overexpress transmembrane receptors. Somatostatin receptors (SSTR) are a family of integral membrane glycoproteins, that have been effectively targeted using SST agonists and antagonists in different cancer types, including neuroblastoma, meningioma, low-grade gliomas and glioblastoma multiforme (GBM). This approach utilizes an octreotide derivative as targeting moiety, coupled to a radionuclide using bifunctional chelators such as DOTA (tetraazacyclododecane-tetra-acetic acid) and DTPA (diethylenetriamine penta-acetic acid). Two radiopeptides most commonly used are [^90^Y]-DOTATOC ([DOTA^0^-D-Phe1, Tyr^3^] octreotide) and [^177^Lu]-DOTATATE ([DOTA^0^-Tyr^3^] octreotate) (Figure 1) [38,39,40]. Heute et al. reported the treatment of 3 grade IV recurrent GBM patients with ^90^Y-DOTATOC. [^68^Ga]-DOTATOC PET was used prior to administration of the therapeutic compound in order to determine expression of somatostatin receptor 2. The radiopharmaceutical was locally administered via a subcutaneous reservoir system implanted into the resection cavity. MRI and PET confirmed complete remission in one patient and partial remission in the other patients, with only minor side effects. Furthermore, patients experienced a substantial improvement in their quality of life. Whole-body dosimetry showed that tumor cells received a 2-fold higher dose from a single cycle of locally given [^90^Y]-DOTATOC than that of standard whole-brain radiotherapy, while sparing the surrounding brain tissue [41]. 

In an extended pilot study by Schumacher et al. 10 glioma patients (WHO grades II and III; 5 progressive gliomas, 5 extensively debulked) received local administration of varying fractions of [^90^Y]-DOTATOC, either in the bulk tumor or into the resection cavity. For the five progressive glioma patients, PRRT was the only applied modality to counter tumor progression. Remarkably, tumor progression was halted in all cases for 13–45 months in all five patients, and one solid, primarily inoperable anaplastic astrocytoma slowly regressed into a resectable multicystic lesion 2 years after PRRT. PRRT was well tolerated and not associated with intermediate- to long-term toxicity [23,42]. Currently, there are 1 Phase I and 2 Phase II clinical trials ongoing to investigate safety, adverse effects, maximum tolerated dose and efficacy of [^90^Y]-DOTATOC.

#### 4.1.2. Neurokinin Type-1 Receptor

Grade II–IV gliomas have been shown to overexpress the transmembraneous neurokinin type-1 receptor (NK-1). NK-1 receptors are also present on infiltrating tumor cells in the intra- and peritumoral vasculature, which makes it an attractive marker for targeted therapy of the bulk tumor, as well as the infiltrative areas. Substance P (SP) is a neuropeptide, acting as a neurotransmitter and physiological ligand of NK-1 receptors. This 1.8 kDa peptidic vector (Figure 2) is stable at the target site for several hours, but is rapidly degraded within minutes by serum peptidases once it enters blood circulation [43,44]. SP can easily be radiolabeled on its Arg^1^ using chelators DOTAGA (1,4,7,10-tetraazacyclododecane-1-glutaric acid-4,7,10-triacetic acid) or DOTA without loss of affinity. 

14 GBM and 6 glioma patients of WHO grades 2 to 3—all with different treatment regimens preceding PRRT—were administered either [^213^Bi]- or [^90^Y]-DOTAGA-SP intracavitary or intratumorally via a trans-cerebellar catheter. In case of critical tumor location, the less energetic [^177^Lu]-DOTAGA-SP was used to spare healthy tissue. Biodistribution and short-term and long-term toxicity were examined. Not only was the treatment well tolerated, disease stabilization and/or improved neurologic status was observed in 13 of 20 patients and neurologic function improved in 5 of 14 GBM patients within 2 weeks after PRRT. Patients that underwent debulking surgery after intratumoral PRRT benefitted from better tumor margin demarcation and reduced intraoperative bleeding due to the radiation effects on tumor vasculature. 

Building on the promising results with [^90^Y]-DOTA-SP, Cordier et al. and Królicki et al. injected 5 and 20 patients resp. with inoperable grade II-IV gliomas intratumorally with [^213^Bi]-DOTA-[Thi^8^, Met(O_2_)^11^]-SP as the primary therapeutic modality. They confirmed the absence of significant local or systemic toxicity, and high retention of the radiopharmaceutical at the tumor site. Tumors showed radiation-induced necrosis and demarcation of the tumor margin, which was validated during resection. Patients who benefitted the most from [^213^Bi]-labeled substance P were the ones with relatively small tumors in the study (12.0–17.1 cm^3^), where adequate intratumoral dose-distribution could be achieved [44,45]. A 10-year follow-up study by the Cordier group, examining long-term effectiveness and safety, showed that 2 out of 5 patients are still alive, with no apparent signs of tumor recurrence and without new functional deficits [46]. A randomized, controlled phase II study is being conducted to further explore the curative effect of this approach [47]. 

In order to compensate for the long diffusion time of SP and the short half-life of ^213^Bi, [^225^Ac]-DOTA-SP was administered intratumorally or into the post-surgical cavity to 21 patients with recurrent grade II-IV glioma after standard treatment. [^68^Ga]-DOTA-SP was co-injected to assess biodistribution by PET-CT. Local administration of [^225^Ac]-DOTA-SP was well tolerated, with mild, transient observations of edema, aphasia or epileptic seizures. OS ranged from 4 to 32 months since primary diagnosis [48]. Patient recruitment for dose escalation studies is ongoing.

#### 4.1.3. Prostate Membrane Antigen

Prostate membrane antigen (PSMA) is a type II transmembrane protein that is highly expressed in poorly differentiated, metastatic, and castration-resistant prostate cancer. Vipivotide tetraxetan (PSMA-617) is a high-affinity prostate-specific membrane antigen (PSMA) inhibitor (Figure 3) that has recently gained much attention in relation to targeting of prostate cancer metastasis. PSMA-617 shows fast tumor uptake, a high internalization rate, extended tumor retention and rapid clearance of unbound ligand, which makes it an ideal theranostic tracer. Case reports of whole body PET/CT scans using [^68^Ga]-labeled PSMA-617 revealed tracer accumulation in intracranial lesions [49,50,51] apart from the primary tumor site. After confirmation of [^68^Ga]-PSMA-617 brain tumor uptake in PET/CT, Wei et al. treated 2 patients with castration-resistant prostate cancer (CRPC) patients with cerebral metastasis with 3–4 cycles of [^177^Lu]-PSMA-617, combined with local radiotherapy. After the combined therapy, all metastases, particularly the cerebral lesions, showed significant decrease in size and PSMA expression [52]. Targeted α-therapy with [^225^Ac]-PSMA-617 has also been explored by Kratochwil et al. in 40 CRPC patients as a salvage therapy in comparison to [^177^Lu]-PSMA-617. Anti-tumoral efficacy was higher for [^225^Ac]-PSMA TAT than for [^177^Lu]-PSMA TRNT, but toxicity to the salivary glands was also increased, leading to severe xerostomia (dry mouth syndrome) as the dose-limiting toxicity. For advanced stage patients an 8-weekly regime of 100 kBq/kg [^225^Ac]-PSMA-617 per cycle presented a reasonable compromise between toxicity and clinical response. These findings indicate however that further modifications of the treatment regimen with regard to side effects might be necessary to further enhance the therapeutic range [53,54]. 

Even though the brain is an uncommonly involved organ in prostate cancer metastasis—most frequently bone and soft tissue are the sites of dissemination - increased PSMA expression has also been observed in neovasculature of high grade (33.3%) and low grade (8.3%) gliomas [55]. Similarly, Wernicke et al. showed in 14 patients with brain metastatic breast cancer that in all cases the tumor-associated vasculature was highly PSMA-positive [56]. These observations add to the potential clinical applicability of PSMA as a molecular target for intracranial TRNT.

### 4.2. Radioimmunotherapy

Radioimmunotherapy (RIT) utilizes a monoclonal antibody radiolabeled typically with a β^−^-emitting radionuclide for the tracking and elimination of cancer cells. At present, only one radioimmunoconjugate (RIC)—[^90^Y]-Ibritumomab tiuxetan (Zevalin^®^)—is approved by the Food and Drug Administration (FDA) and used to treat indolent CD20-positive B-cell lymphoma. Data from clinical application shows that a single treatment cycle with RICs can induce the same degree of anti-tumor efficacy as multiple cycles of conventional chemotherapy, with a far less severe toxicity profile [57]. In contrary to hematological cancers, such as lymphoma and leukemia, progress in the field of solid tumors has been slow. This is partly due to their poor tissue permeating power and high dose needed for bulk tumors. For this reason, RICs have been investigated in the context of combating micrometastatic lesions or minimal residual disease. Based on published clinical trials, over 500 patients have been treated worldwide with RICs targeting a multitude of receptors and coupled to various α- and β^−^-emitting radionuclides, administered directly into the resection cavity after surgical debulking.

#### 4.2.1. Epidermal Growth Factor Receptor 

Epidermal Growth Factor Receptor (EGFR) is a receptor tyrosine kinase that is found to be overexpressed on the cell surfaces of various solid tumors. EGFR activation in GBM promotes cellular proliferation via the MAPK and PI3K–Akt pathways. EGFR overexpression correlates with increased tumor growth rate, invasiveness and decreased survival, which makes it a favorable target for TRNT of GBM [58,59].

Various clinical trials report the use of the murine anti-EGFR mAb 425 antibody, radiolabeled with ^125^I via by the Iodogen method. Since EGFR is an internalizing receptor, an Auger-emitter like ^125^I will guarantee the most specific targeting potential of commonly used radionuclides. Intravenous administration of [^125^I]-mAb 425 either alone or with standard of care treatment significantly improved median survival [60,61,62]. In the largest Phase II trial to date, including 192 patients, combination treatment of [^125^I]-mAb 425 and TMZ provided the greatest survival benefit with a median survival of 20.4 months, compared to treatment of [^125^I]-mAb 425 alone, which was 14.5 months). Interestingly, compared to most clinical trials concerning TRNT in the brain, [^125^I]-mAb 425 was not administered locally into the resection cavity, but rather through an intravenous injection. This highlights the efficacy of this treatment considering systemic distribution of the radiopharmaceutical. Importantly, apart from skin irritation at the injection site no other toxicities were attributed to the intravenous injection [61]. 

A small-scale Phase I single-dose study has been performed using [^188^Re]-labeled Nimotuzumab, a humanized anti-EGFR antibody. ^188^Re is a short-lived β^−^-emitting radionuclide and can be directly conjugated to antibodies as a glucoheptonate complex. ^188^Re is easily obtained from an in-house ^188^W/^188^Re generator that is similar to the ^99^Mo/^99m^Tc generator, making it very convenient for clinical use. [^188^Re]-labeled Nimotuzumab was administered post-surgery into the resection cavity of 3 patients with anaplastic astrocytoma (AA) and 8 patients with GBM. 1 GBM patient died in progression 6 months after TRNT, 1 GBM and 1 AA patient developed stable disease within 3 months. One GBM patient had partial response for more than 1 year and 2 patients (1 GBM and 1 AA) were asymptomatic and in complete response after 3 years of treatment. Dose-dependent neurotoxicity was observed in 2 treated patients [24].

#### 4.2.2. Epidermal Growth Factor Receptor Mutant Variant III

Aberrant amplification of EGFR often coincides with deletion, or mutation of at least one receptor tyrosine kinase, which is correlated with tyrosine kinase inhibitor therapy resistance. In addition, around 50% of patients with EGFR amplification harbor a specific mutation—known as EGFRvIII—which results in the deletion of part of the extracellular domain and constitutive, ligand-independent activation of the EGFR’s signaling pathways [63,64]. Since EGFRvIII is tumor-specific, a known tumor-driver, and resistant towards several EGFR-directed therapeutics, it is a perfect target for TRNT. 

Several antibodies generated against wildtype EGFR, such as Cetuximab and mAb 425 also recognize EGFRvIII. However, applying them via systemic administration could lead to systemic toxicity due to the expression of wildtype EGFR on healthy tissues. Despite the generation of EGFRvIII-specific antibodies [65,66,67], implementing them as TRNT vehicles has not reached clinical examination yet.

#### 4.2.3. DNA-Histone H1 Complex

Peregrine Pharmaceuticals, Inc. (Tustin, CA, USA) introduced a [^131^I]-labeled chimeric monoclonal antibody ([^131^I]-chTNT-1/B MAb)—commercially known as Cotara^®^—specific for a universal intracellular antigen, namely histone H1 complexed to DNA. In patients with high-grade glioma and GBM, this protein is exposed in high quantities within the necrotic core. Cotara^®^ does not penetrate healthy cells with an intact cell membrane. However, in the necrotic regions of malignant solid tumors, this abundant, non-diffusible protein provides an anchor for Cotara^®^, where it delivers a cytotoxic dose of ^131^I radiation to the adjacent living tumor cells [68]. 

By administering Cotara^®^ through convection-enhanced delivery, not only the bulk tumor, but also the invasive glial cells disseminated into normal brain tissue can be irradiated. 51 high-grade glioma patients have been treated in Phase I/II clinical trials to date. Treatment-related adverse effects included brain edema (16%), hemiparesis (14%) and headache (14%) but were mostly reversible by corticosteroids. 

Although the Phase II trial indicated therapeutic efficacy, an insufficient number of patients were included for treatment with a consistent dose [68,69,70]. 

#### 4.2.4. Tenascin

Tenascin is an extracellular matrix glycoprotein that is expressed ubiquitously in several cancer types, including high-grade gliomas, but not in normal brain tissue [71]. Riva et al. exploited this tumor-specificity in 1995 by locally treating 50 recurrent and newly diagnosed glioma patients with 2 murine monoclonal antibodies, BC-2 and BC-4, radiolabeled with ^131^I using the Pierce Iodogen method. They observed 3 complete remissions, 6 partial remissions, and 11 patients showing no evidence of disease, corresponding with a response rate of 40%. The local administration of [^131^I]-labeled antibodies did not induce systemic or cerebral adverse effects. Human anti-mouse antibody responses (HAMA) were observed in many cases but did not affect tumor targeting with repeated treatments [72]. 

The Duke Comprehensive Cancer Center has completed 5 Phase I/II clinical trials using the radiolabeled anti-tenascin murine monoclonal antibody 81C6. Promising results have been obtained in many patients, with median survivals of up to 22 months seen in GBM patients treated with [^131^I]- and [^211^At]-labeled 81C6 compared with approximately 11-month survival for conventional treatments. Labeling of 81C6 was performed by either synthesizing N-succinimidyl 3-[^211^At]astatobenzoate before reacting with the antibody [22] or radioiodination through modified Pierce Iodogen procedure [73]. Despite the occurrence of HAMA responses in 79% of patients, no adverse effects related to HAMA were observed. Toxicity was mostly confined to reversible hematologic and neurologic events. Some patients however showed significant signs of both hematologic and neurologic toxicity and further clinical development. The trial concerning the astatinated antibody demonstrated transient neurotoxicity, although no dose-limiting toxicity was reported [22,73,74,75,76]. Orphan designation of ^131^I—81C6 was granted in 2006 in the United States for the treatment of primary brain tumors. However, demonstration of quality, safety and efficacy is still necessary before this iodinated antibody can be granted marketing authorization. Further clinical examination has not been reported with the iodinated 81C6 antibody since.

In order to maximize the targeting potential of monoclonal antibodies while minimizing the toxicity of long-circulating RICs, a 3-step pretargeting strategy has been investigated in which the targeting antibody is administered first, followed by the injection of a radiolabeled low molecular weight tracer. For the targeting of tenascin, the biotin-coupled BC-4 antibody was administered first, followed 24 h later by avidin, and finally, after an additional 18 h a [^90^Y]-labeled biotin was administered. The efficacy of this approach was validated in 73 patients with histologically confirmed GBM, treated with the 3-step [^90^Y]-biotin based RIT. Stabilization of disease was achieved in 75% of patients, while 25% progressed. In patients treated with RIT alone, median OS was 17.5 months, while in patients treated with a combination regime of RIT + TMZ OS was 25 months, with neurologic toxicity being the dose-limiting factor [77].

#### 4.2.5. Fibronectin

Angiogenesis is the formation of new blood vessels and is a common feature of solid tumors to provide them with oxygen and nutrients. It is regulated by a number of cell surface receptors and extracellular matrix adhesion molecules. Angiogenic modulators can be useful markers for the delivery of therapeutic agents to the tumor site. Fibronectin can be expressed as several isoforms depending on alternative splicing, of which 3 are found on tumorous tissue, namely IIICS, extra domain (ED) A and ED-B [78]. Starting from a human antibody targeting the ED-B, termed ‘L19’, a ~80 kDa construct was generated consisting of 2 single-chain variable fragments fused by a CH4 domain of the human IgE [79]. This construct, L19SIP, was radioiodinated using a modified chloramine-T method [80] and has been clinically investigated in several solid tumor types after intravenous administration [81], including metastatic solid brain tumors. [^124^I]-labeled L19SIP has been used by Poli et al. in immune-PET imaging to predict the delivered dose to metastatic lesions and healthy organs in patients with brain metastases from solid tumors. They showed that the delivered dose was <2 Gy to the bone red marrow, which is the main dose-limiting organ for this type of therapy. Intracranial lesions received between 0.73 and 8.15 Gy, depending on the size of the lesion. Apart from one metastatic lesion in the liver (35.84 Gy), these values are comparable to those obtained in extracranial metastatic lesions [82]. Patients eligible for RIT (target/background ratio of brain metastases >4 and bone marrow dose <2 Gy) were then treated with [^131^I]-L19SIP. Preliminary results show that in 3 out of 4 patients, intra- and extracranial lesions showed reduced [^18^F]-FDG-uptake during the 6-month follow-up. The reduction of glucose metabolism in the lesions suggests the potential clinical efficacy of [^131^I]-L19SIP-mediated RIT [83].

## 5. Preclinical Validation

In order to guarantee a successful bench-to-bedside translation of radiopharmaceuticals, treatment optimization in relevant preclinical models is essential. Subcutaneous or orthotopic xenografts of overexpressing cancer cells in rodents are useful to determine tracer-specificity, but poorly reflect the heterogeneity and complexity of tumor progression. Preclinical models often lack certain GBM features such as single-cell invasion, tumor necrosis, or microvascular proliferation [84]. Preclinical models for brain metastasis are often generated in the same fashion as primary brain tumors, i.e., by stereotaxic intracranial injection of tumor cells. This single-cell depository inadequately mimics the natural process of metastasis to the brain and tumor-microenvironment adaptation. A breakthrough in metastatic disease research was the development of a brain-seeking (MDA-231BR) clone by Yoneda et al. of the human breast cancer cell line MDA-MB-231 by repeated sequential passages in nude mice of intracardially injected cells obtained from brain metastases [85]. 

Preclinical validation of a targeted approach concerning human biomarkers or patient-derived xenografts also requires the use of immune-deficient mice lacking a full immune system that cannot be activated against the tumor. Naturally occurring GBM rodent models are often unpredictable regarding tumor growth or do not express the biomarker of interest. Large animals with naturally occurring GBM or brain metastasis such as dogs and pigs are scarce, and treatment costs are in the same range as those for humans [86]. 

The molecular signature for cancer cells with increased capacity for brain involvement includes expression HER2, EGFR, heparanase and Notch1 expression [87]. Expression of these markers on the primary tumor might indicate greater risk of metastasis and warrant targeted therapeutic approaches. In a preclinical model of Trastuzumab-resistant HER2-positive carcinomatous meningitis, [^211^At]-labeled Trastuzumab was administered intrathecally, thereby bypassing the BBB. ^211^At is due to its short radiation range and short half-life of 7.2 h ideal to minimize radiation dose to the spinal cord or systemic toxicity after protein resorption from the cerebrospinal fluid into the general circulation. Treatment with [^211^At]-labeled Trastuzumab led to a doubled median survival compared with untreated controls as well as a group of animals receiving unlabeled trastuzumab [88]. 

The efficacy of Trastuzumab was compared to a radioiodinated anti-HER2 heavy-chain-only antibody fragment [89] in an intracranially inoculated HER2-transduced MDA-MB-231Br mouse model. Treatment with therapeutic doses of the [^131^I]-SGMIB-anti-HER2 VHH1 significantly increased survival compared to Trastuzumab [90]. Clinical translation of the [^68^Ga]-DOTA-anti-HER2 VHH1 for PET imaging of HER2-positive brain lesions is ongoing [91,92] and the theranostic value of [^131^I]-SGMIB-anti-HER2 VHH1 for detection and treatment of breast cancer brain metastasis is currently under investigation [93].

Once a glial tumor occurs, or when circulating metastatic cancer cells nest within the brain, tumor growth is critically dependent on angiogenesis. Thus, therapies targeting markers involved in this process could prove beneficial in the management of brain tumors and metastases. Destruction of neovasculature can hinder tumor progression, as well as enhance drug penetration into the tumor’s interstitial space [94]. Since destruction of the tumoral vascular network can also restrict the delivery of oxygen, leading to intratumoral hypoxia, the use of α-emitting radionuclide targeted therapies towards the tumor vasculature could be warranted [95,96,97,98]. Falzone et al. used an in silico model of early brain metastasis to evaluate the efficacy of several α- and β^−^- and Auger-emitting radionuclides in the context of vascular cell adhesion molecule 1 (VCAM-1) targeting antibodies. VCAM-1 has been found to play a pivotal role in adhesion of cancer cells to the vascular endothelium and mediates specific tumor-stromal interactions, leading to metastasis of VCAM-1-expressing cells to lung, bone and brain [99]. Monte Carlo simulations showed that the anti-VCAM-1 antibody coupled to the theranostic α-emitting ^212^Pb radionuclide showed the most favorable dose profile and relative biological effectiveness [100]. 

Behling et al. reported the first preclinical investigation of targeted α-particle anti-vascular therapy in a transgenic GBM mouse model. They validated the therapeutic efficacy of the anti-vascular endothelial (VE)-cadherin antibody E4G10, radiolabeled with ^225^Ac, compared to a radiolabeled isotype-matched control antibody. VE-cadherin is an essential adhesion glycoprotein for proper vascular development and maintenance. Blocking of VE-cadherin interactions leads to inhibition of tumor angiogenesis and growth [101]. They observed an absorbed dose in the tumor that was 7.7 times higher than that of a normal brain, and a significant decrease in tumor growth compared to the control group. The median survival in the [^225^Ac]-E4G10 group was 21 days compared to 9 days in the control group [97]. 

The DNA repair enzyme poly(ADP-ribose) polymerase 1 (PARP-1) is overexpressed in GBM in order to maintain genomic integrity through an accelerated cell cycle. Since PARP-1 is generally overexpressed in highly proliferative cancer cells, expression in healthy neuronal tissue is low [102]. Exploiting the success of several PARP-1 inhibitors (PARPi), Jannetti et al. synthetized a radioiodinated PARPi and administered this probe locally in orthotopic brain tumors via CED. Dosimetry based on SPECT/CT data indicated that 9.1 Gy were delivered to the whole brain of tumor-bearing animals, whereas less than 1 Gy was delivered to healthy animals, showing that in a preclinical setting CED-delivered [^131^I]-PARPi specifically accumulates in cancer cells, while being quickly cleared from healthy tissue. Further experiments will need to show whether [^131^I]-PARPi has a beneficial therapeutic efficacy over the existing standard of care [103].

## 6. Discussion

Historically, brain tumors have been the metaphorical ‘problem child’ in the world of cancer therapy. As the usual standard of care proves insufficient to guarantee good survival odds, treatments and better outcomes for primary brain tumors and metastases have long lagged behind those of other tumors. The invasive nature of glial tumors and protective environment of the CNS require more extreme measures to eradicate residual cells that can give rise to relapse. However, radical therapeutic interventions and the preservation of neurological functions are opposing goals.

Identification of patients’ genetic and phenotypic tumor profile can help predict therapeutic responses, risk stratification, and alter treatment selection. The discoveries regarding tumor-specific biomarkers have led to an array of novel therapy options in the field of immunotherapy, fluorescence-guided surgery and targeted therapies. More than ever it becomes clear that future treatment will have to consist of combination regimens to achieve a curative effect. 

The current challenge is to translate preclinical findings into clinically applicable therapies. Very few trials that focus on CNS disease have progressed beyond phase II, and historically most clinical trials have excluded patients once CNS metastasis occurs [104]. A recurring problem has been the recruitment of sufficient patients for clinical trials, where low statistical power can complicate interpreting trial outcomes and justify further phase development [105]. Since brain tumors are highly aggressive, the few patients that present this uncommon disease often don’t see benefit in undergoing an additional treatment regime with potentially more adverse events, while therapeutic success is not guaranteed. Additionally, GBM is classified as an orphan disease, and its rarity constitutes a stumbling stone for many funding agencies, making innovative clinical advancement slow and frustrating.

Currently, diagnosis of brain metastases is performed based on MRI, CT or PET and is often treated with the same regime as the primary tumor, or at least regarded as having the same phenotypical properties as the primary tumor. However, brain metastases often present with different mutations from their systemic primary tumors. The growing knowledge of brain metastasis biology has led to the development of new targeted therapies with the focus on CNS bioavailability. Molecular imaging can aid in identifying target-presence and therapy response. MRI or CT can often be misleading due to pseudoprogression (therapy-induced edema that mimics tumor progression), or to pseudoresponse (reduced tumor-enhancement on MRI caused by corticosteroids or altered vessel permeability to contrast agents caused by anti-angiogenic drugs). There is a growing interest in applying the same or comparable targeting moiety coupled to either a diagnostic or a therapeutic radionuclide, such as ^68^Ga and ^90^Y resp. Alternatively, targeting moieties can be labeled to inherently therapeutic radionuclides that also emit γ-radiation in their decay scheme, such ^177^Lu and ^131^I, that can be used for imaging and treatment response. 

Preclinical research into brain tumors and metastases often utilizes homogenous cell populations overexpressing the biomarker of interest. This is a poor representation of the clinical situation where lesions are heterogeneous in cell type, receptor expression, BBB integrity and vascularization. If targeted therapies are efficient in only eliminating target-expressing cells, tumor heterogeneity will lead clonal expansion of receptor-negative cells. The hallmark of β^−^-emitting radionuclides is their ability to irradiate receptor-negative cells due to their long travel path. Induction of the bystander effect and the formation of ROS also aids in eliminating not only targeted cells, but also tumor cells in the vicinity of the targeted cells. Beta-RIT for brain tumors has been explored in the late 1990s using [^131^I]- or [^90^Y]-labeled to anti-tenascin antibodies. Due to the limited number of patients and the advanced cancer stages, clinical responses in these trials were difficult to interpret. While the potential danger of beta-radiation is the exposure of healthy brain tissue, no to minimal neurotoxicity was observed in the use of β^−^-emitting radionuclides.

The fact that recurrent malignant gliomas and brain metastases clinically manifest as multifocal lesions, underscores the need for locally-active therapies as an essential part of glioma treatment. While beta-RIT certainly has been proven to be an efficient therapeutic option, TAT has important physical and radiobiological advantages. Considering the delicate nature of neural tissues and hypoxic tumor environment, the short range and oxygenation-status independency of α-emitting radionuclides could lead to a safer, yet more efficient cell killing concerning brain malignancies. For invasive tumor cells beyond visible tumor margins TAT may become the cornerstone in precision medicine. 

Despite these advantages, (pre)clinical investigation of TAT has only recently been gaining more attention, with the improved availability of clinically applicable isotopes. Isotopes such as ^211^At require production by a high-energy accelerator, which are very limited in number and its short half-life requires on-site production of radiopharmaceuticals. The introduction of an easy-to-use ^225^Ac/^213^Bi generator has facilitated the (pre)clinical application of these isotopes. However, despite the ~10-day half-life of ^225^Ac allowing centralized production, generator availability remains limited, and a commercially viable production source is still not available. Also, the high therapeutic doses needed for clinical translation, combined with the short half-life of ^213^Bi constitutes a real challenge for trial design.

Restricted isotope availability and complex conjugation chemistry severely limit research and development opportunities in TAT, and clinical implementation thereof. Another consideration that must be made is the strong recoil caused by alpha decay. Recoil daughter isotopes can escape from their conjugation agent and lead to systemic toxicity when not retained at the tumor site. This is particularly of interest for ^225^Ac, which emits up to 4 α-particles during its decay to stable ^209^Bi. Other limitations that hinder translation to clinical use are the costs due to strict toxicology regulations enforced by the FDA. Evaluating a new radiopharmaceutical for use in humans is often beyond the budgets of academic institutions and are a major regulatory impediment to translation into clinical practice.

## Figures and Tables

**Figure 1 pharmaceutics-11-00376-f001:**
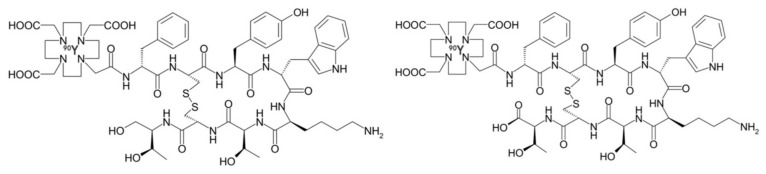
Structural formula of [^90^Y]-DOTATOC (**left**) and [^90^Y]-DOTATATE (**right**). These are the most commonly used SST analogs in nuclear medicine.

**Figure 2 pharmaceutics-11-00376-f002:**
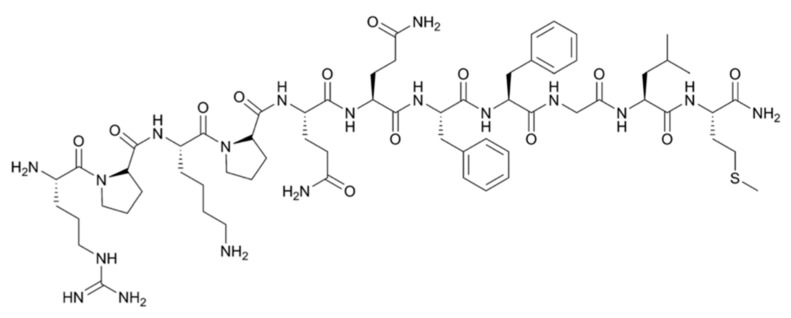
Structural formula of the 11-amino acid neuropeptide Substance P.

**Figure 3 pharmaceutics-11-00376-f003:**
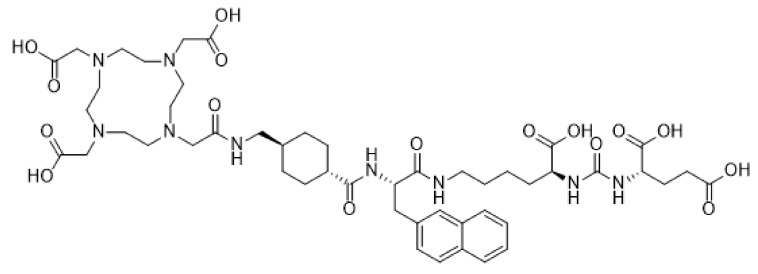
Structural formula of the PSMA inhibitor Vipivotide tetraxetan (PSMA-617).

**Table 1 pharmaceutics-11-00376-t001:** Commonly used therapeutic radionuclides and their radiochemical characteristics.

Radionuclide	Abbreviation	Emission	Half-Life	Energy_max_ (keV)	Travel Distance	Characteristics of Radiation Class
Actinium-225	^225^Ac	Alpha/beta^−^/gamma	9.92 days	7.069	50–100 μm	+ Short range, high energy
Astatine-211	^211^At	Alpha	7.20 h	5.867	50–100 μm	+ Double stranded DNA breakage
Bismuth-213	^213^Bi	Alpha/gamma	46 min	6.051	50–100 μm	+ Oxygen independent
Lead-212	^212^Pb	Alpha/beta^−^/gamma	10.64 h	8.785	50–100 μm	− No crossfire
Iodine-131	^131^I	Beta^−^/gamma	8.02 days	606	200 µm–1 mm	+ Crossfire effect
Lutetium-177	^177^Lu	Beta^−^/gamma	6.68 days	498	230 µm	− Oxygen dependent
Rhenium-188	^188^Re	Beta^−^/gamma	16.98 h	2.110	11 mm	− Long range, low energy
Yttrium-90	^90^Y	Beta^−^	2.67 days	2.280	12 mm	− Single stranded DNA breakage
Indium-111	^111^In	Auger/gamma	2.8 days	245	4 nm	+ Very short range
Iodine-125	^125^I	Auger/gamma	59.49 days	35	2 nm	− Necessary to be internalized

**Table 2 pharmaceutics-11-00376-t002:** Overview of clinical advancements in Targeted Radionuclide Therapy of brain tumors and metastases.

Disease	Target	Compound	Administration Route	Testing Phase	Results	Reference/Clinical Trial Identifier:
Neuroblastoma, meningioma, glioma, GBM	Somatostatin receptors	[90Y]-DOTATOC	Directly injected or via subcutaneous reservoir system into resection cavity	Phase I/II	+ Partial or complete remission. − Well tolerated, with minimal neurological toxicity.	[23,38,39,40,41,42] NCT03273712, NCT00006368, NCT02441088, NCT00006368
Grade II–IV gliomas	Neurokinin type-1 receptor	[90Y]-DOTAGA-Substance P	Intratumorally via trans-cerebellar catheter	Phase I	+ Disease stabilization and/or improved neurologic status. − No significant local or systemic toxicity	[43]
		[225Ac]-DOTA-Substance P	Intratumorally or into the post-surgical cavity	Phase I/II	+ OS prolongation up to 32 months − Well tolerated, with mild, transient edema, aphasia or epileptic seizures.	[48]
		[213Bi]-DOTA-Substance P	Trans-cerebellar catheter	Phase I/II	+ Partial or complete remission. Disease stabilization and/or improved neurologic status. − No significant local or systemic toxicity	[43,44,45,46,47]
Grade II–IV gliomas, anaplastic astrocytoma	Epidermal growth factor receptor (mutant variant III)	[125I]-mAb 425	Intravenous	Phase II	+ Median survival benefit of 20.4 months − Mild skin irritation at injection site	[60,61,62]
		[188Re]-labeled Nimotuzumab	Directly injected into resection cavity	Phase I	+ 1/11 partial response, 2/11 complete response after 3 years. − Dose-dependent neurotoxicity was observed in some patients	[24]
Grade II–IV gliomas, anaplastic astrocytoma	DNA-histone H1 complex	[131I]-chTNT-1/B MAb	Intratumorally via convection-enhanced delivery	Phase I/II	+ Clinical efficacy not definitively established due to low patient number. Median survival time was noted as 37.9. weeks for subset of patients. − Edema, hemiparesis and headache	[68,69,70] NCT00677716, NCT00509301, NCT00128635, NCT00004017
Grade I–IV gliomas	Tenascin	[131I]-BC-2 mAb or [131I]-BC-4 mAb	Intratumorally	Phase I/II	+ Partial or complete remission. Response rate of 40% − No systemic or cerebral adverse effects, HAMA response did not affect tumor-targeting	[72]
		3-step pretargeting strategy with biotin-coupled BC-4 + Avidin + [90Y]-Biotin	Trans-cerebellar catheter	Phase I/II	+ Disease stabilization in 75% of patients. OS prolonged to 17.5 and 25 months with TRNT alone or TRNT+TMZ resp. − Transient hematological toxicity, mild allergic reaction	[77]
		[131I]- or [211At]-labeled 81C6 mAb	Directly injected into resection cavity	Phase I/II	+ Median survivals of up to 22 months − Reversible hematologic and neurologic events. No adverse effects related to HAMA response	[22,73,74,75,76] NCT00003461, NCT00003484, NCT00002752, NCT00003478, NCT00002753
Brain metastasis from breast carcinoma and NSCLC	Fibronectin	[131I]-L19SIP	Intravenous	Phase I/II	+ Intra- and extracranial lesions showed reduced [18F]-FDG-uptake during 6-month follow-up − No adverse events reported	[81,82,83] NCT01125085
Brain metastasis from prostate cancer	Prostate membrane antigen	[177Lu]-PSMA-617	Intravenous	Phase I	+ Significant decrease in size and PSMA expression − Minimal toxicity to salivary glands	[52] NCT03511664
		[225Ac]-PSMA-617	Intravenous	Phase I/II	+ More potent effect than [177Lu]-PSMA-617 − Severe xerostomia	[53,54]

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
