# Peer review of "Beyond the Barrier: Targeted Radionuclide Therapy in Brain Tumors and Metastases"

_pharmaceutics, 2019, doi:10.3390/pharmaceutics11080376_

Round 1
Reviewer 1 Report
Puttermans and colleagues, in this narrative review, present the current knowledge on the targeted radionuclide techniques of primary and metastatic brain cancers. The began by highlighting the challenges with the management of malignancies of brain and the dismal outcome with the current standard of care treatment options. The promising aspect of targeted radionuclide therapy was presented. The authors also highlighted the molecular targeted that have been exploited for targeted radionuclide therapy.
The review is well-written and summarizes the current knowledge on this subject. The subject addressed is topical and will be of interest to the readers.
A few comments are as follows:
· Iodine-125 is an emitter of auger electron that has been used for TRNT. It is however not included in table 1.
· “Theoretically, a cancer cell can be killed a single hit of an -particle, and its efficacy…..” Alpha particle causes double-stranded DNA damage that is difficult to repair and hence one hit is capable of killing the cell. I will suggest the author exhibit some caution and rather say a few hits can cause tumor kill.
· Another important point regarding targeted alpha therapy is cell killing independent of cell cycle phase making slowly dividing tumor to be killed to the same extent as rapidly dividing cells.
· The force of recoil exhibited by an alpha particle emission is great and can cause bond breakage. This becomes very important in radionuclides such as Ac-225 which emits multiple alpha particle. Bond breakage may lead to release of free radionuclide consequently increasing whole-body radiation dose.
· In many instances, the authors did not provide a reference immediately after referring to a study. Please identify such places and make correction.
· Despite the title of this review, the level of discussion on TRNT of metastatic brain malignancies lacks depth compared with the discussion on primary brain tumors. In the recent times, case reports/series have been published regarding PSMA-targeted radionuclide therapy of prostate cancer metastatic to the brain using Lu-177 and Ac-225. I encourage the authors to include a brief discussion on this potential application of TRNT in brain metastases of prostate cancer.
Reviewer 2 Report
In their review manuscript, Puttemans et al. aim to provide “an overview of targeted radionuclide therapy approaches for the treatment of primary brain tumors and brain metastases, with emphasis on biological targeting moieties that specifically target key biomarkers involved in cancer development and treatment resistance.”
The authors spent much effort assembling 85 references for their manuscript and describe a lot of detail. However, despite the large effort by the authors, the main challenge with this manuscript is that it does not very well condense the many details and does not effectively extract the most pertinent aspects; instead, it mostly is a lengthy listing of individual observations. The use of radionuclides for therapy has been explored for decades: which ones have emerged from this lengthy process as the most effective today? Which ones advanced to routine clinical use?
1. Table 1 represents a useful summary. There should be more tables listing the individual trials and clearly separating them into Phase I, Phase II, etc. This would allow shortening of the main text and would support the presentation of more critical analysis of the current progress and challenges in the field of therapeutic radionuclides.
2. From the many approaches that are presented, it does not become clear which ones are the most promising. Over the past decades, many approaches were tested, but few advanced to clinical practice. Which ones are in clinical use today? Which ones were successful in Phase II or Phase III trials? A summary table would be useful.
To pick an example that requires more detail: On page 4, the authors mention methods to open the blood-brain barrier, such as mannitol, RMP-7, and focused ultrasound. But it is not mentioned which of these have been combined with radionuclide therapy. Nor is it mentioned how risky some of these methods, in particular intracarotid mannitol, can be. The referenced clinical studies (refs. 23, 24) with the use of RMP-7 are 15 years old: is there an update since then?
To pick another example: On page 5, the authors describe pilot studies with Yttrium-90 DOTATOC in glioma patients and mention that this treatment modality “was well tolerated” and “tumor progression was halted in all cases for 13-15 months.” This appears to represent a remarkably encouraging outcome. However, the references that are cited (#16,30) are from 2002 and 1999, meaning that these studies were performed in the last century. The authors should comment why this approach, despite encouraging results, never made it into clinical use with glioma patients. Instead, the currently ongoing clinical trials are with neuroendocrine tumors. Was Yttrium-90 DOTATOC abandoned for gliomas (why)? What was the rationale (scientific observations) to switch to endocrine tumor application?
Another example: On page 8, the authors describe “promising results” obtained in several clinical trials with radiolabeled anti-tenascin antibody 81C6. These results were published over a decade ago. What has happened since? What is the current status of 81C6? Was it abandoned? Why?
The above examples illustrate how the authors should minimize their lengthy listing of individual observations (put it into tables instead), but rather focus on CURRENT successes and remaining challenges in this field, including a critical analysis as to why so many “promising” outcomes appear to have fizzled.
3. (a) The main text should be shortened by 50%. (b) The Introduction should be more specific. For example, the statement “Brain tumors remain … the most lethal of all cancer type” and “diagnosis is still a death sentence”, is not entirely correct, because there are also benign brain tumors. For instance, meningiomas and pituitary adenomas generally do not represent a “death sentence”. Radionuclides are used also for non-glioma tumors. (c) The Discussion section has unnecessary introductory points, some of them commonplace, that should go to the Introduction (or already are mentioned there). It also contains text passages that should go into the main text, because they do not represent critical points of discussion. An interesting point of Discussion is the limitation set by the availability of clinically relevant isotopes, complex conjugation chemistry, and limited number of high-energy accelerators. What is being done to decrease these limitations?
4. Brachytherapy is not mentioned. It would be useful (in the Introduction) to provide a distinction (advantages, drawbacks) between “targeted radionuclide therapy” and the better known and perhaps more widely used “brachytherapy”.
5. In their Abstract, the authors promise “an overview of targeted radionuclide therapy approaches for the treatment of primary brain tumors and brain metastases, with emphasis on biological targeting moieties that specifically target key biomarkers involved in cancer development and treatment resistance.” It does not become clear how the authors have delivered on their “emphasis” on targeting “key biomarkers involved in cancer development and treatment resistance.” The authors should clearly introduce and define “key biomarkers of cancer development” and “key biomarkers of treatment resistance”. It is not explained how the mentioned targets (somatostatin receptor, EGFR, tenascin, etc.) represent biomarkers of treatment resistance.
Reviewer 3 Report
Page 2 line 45: 2. THE PROBLEM OF THE BRAIN. The subtle is not clear and confused. This part is to mention why the brain is a major hurdle to manage the tumors localized within the brain. Therefore, it would be better to have a paragraph to describe the special structure of BBB. Why can it cause the treatment difficult?
Page 2 line 75-78. It is not clear why immune-privilege in the brain can lag immunotherapy. Please define the immunotherapy approach. In that case, why do you mention “This has led to an expanse in development of tyrosine kinase inhibitors, monoclonal antibodies and antibody fragments, nanoparticles, etc.”? Usually, antibody development is considered as immunotherapy.
In Table 1, it would be better to include the columns of "Advantage" and "Disadvantage". That would help the readers understand the different function and potential application.
In addition, it would help to give a brief description of the character of alpha/beta/gamma emitting radionuclides below Table 1.
What is the advantage and disadvantage of targeted radionuclide therapy compared to targeted chemotherapy? What is the benefit for patients? What is the future direction for developing targeted radionuclide therapy? It can be included in the discussion part.
Round 2
Reviewer 2 Report
Modification of Table 1 and addition of new Table 2 is constructive and quite helpful. As well, the current clinical status of some of these radionuclide approaches has been clarified. A brief mention of brachytherapy has been added. Other minor modifications were applied. Overall, the authors provided adequate revisions.